# Prognostic Role of Circulating Tumor Cell Trajectories in Metastatic Colorectal Cancer

**DOI:** 10.3390/cells12081172

**Published:** 2023-04-16

**Authors:** Valentina Magri, Luca Marino, Chiara Nicolazzo, Angela Gradilone, Gianluigi De Renzi, Michela De Meo, Orietta Gandini, Arianna Sabatini, Daniele Santini, Enrico Cortesi, Paola Gazzaniga

**Affiliations:** 1Department of Pathology, Oncology and Radiology, Sapienza University of Rome, 00161 Rome, Italy; 2Department of Mechanical and Aerospace Engineering, Sapienza University of Rome, 00184 Rome, Italy; 3Department of Molecular Medicine, Sapienza University of Rome, 00161 Rome, Italy

**Keywords:** circulating tumor cells, circulating tumor cell trajectories, metastatic colorectal cancer, liquid biopsy, CellSearch^®^, precision medicine, prognostic and predictive biomarkers

## Abstract

Background: A large amount of evidence from clinical studies has demonstrated that circulating tumor cells are strong predictors of outcomes in many cancers. However, the clinical significance of CTC enumeration in metastatic colorectal cancer is still questioned. The aim of this study was to evaluate the clinical value of CTC dynamics in mCRC patients receiving first-line treatments. Materials and methods: Serial CTC data from 218 patients were used to identify CTC trajectory patterns during the course of treatment. CTCs were evaluated at baseline, at a first-time point check and at the radiological progression of the disease. CTC dynamics were correlated with clinical endpoints. Results: Using a cut-off of ≥1 CTC/7.5 mL, four prognostic trajectories were outlined. The best prognosis was obtained for patients with no evidence of CTCs at any timepoints, with a significant difference compared to all other groups. Lower PFS and OS were recognized in group 4 (CTCs always positive) at 7 and 16 months, respectively. Conclusions: We confirmed the clinical value of CTC positivity, even with only one cell detected. CTC trajectories are better prognostic indicators than CTC enumeration at baseline. The reported prognostic groups might help to improve risk stratification, providing potential biomarkers to monitor first-line treatments.

## 1. Introduction

The ability of invasive tumors to release cancer cells (circulating tumor cells—TCs) in biological fluids, mainly blood, is dramatically important for prognostic purposes in cancer patients. A liquid biopsy allows one to collect and analyze these materials directly using peripheral venous blood sampling, with the great advantage of non-invasiveness and quickness, providing supplemental information from a perspective of precision medicine [1]. The presence of CTCs reflects a very early stage of metastasis spreading and allows for correlating their quantity to prognosis [2]. Furthermore, the serial analysis of CTCs can track cancer evolution during the course of treatment, allowing early detection of the occurrence of drug resistance and monitoring anticancer drug efficacy [3]. Despite this great potential, clinical applications of CTCs are currently limited to breast, colon, and prostate cancer in metastatic settings [4].

CellSearch^®^ (Menarini Silicon Biosystems, Castel Maggiore, Bo, Italy) is the first and only FDA-approved CTC test for cancer patients. In metastatic breast and prostate cancers, the presence of greater than five cells per 7.5 mL of blood was found to be an independent predictor of worse progression-free survival (PFS) and overall survival (OS) [5,6].

Conversely, a validation study for the use of CellSearch^®^ in metastatic colorectal cancer (mCRC) [7] identified a lower cut-off value (≥three CTCs/7.5 mL) as an independent prognostic factor in terms of PFS and OS, recognizing a first peculiarity in the use of this approach in these tumors compared to other cancer types [8]. The explanation for the lower prevalence of CTCs in colorectal cancer patients compared to other cancer types is not univocal, although it has been sometimes ascribed to the high number of CTCs with epithelial-to-mesenchymal transition [8,9].

Furthermore, whereas for breast and prostate cancers, the prognostic role of CTCs based on the specific cut-off clearly proved to be useful for clinical purposes [5,6], evidence has indicated that for mCRC, the presence of even one CTC at baseline is predictive for poor prognosis, suggesting that patients with one–two CTCs should be switched from the favorable to the unfavorable prognostic group [10]. Consequently, the specific CTC cut-off value to be used in mCRC is still debated.

Recently, the analysis of serial CTCs was demonstrated to allow prognostic stratification in metastatic breast cancer [11]. Most authors used CTC values at baseline and at pre-defined time intervals to define different CTC trajectories [8,9].

To date, little is known about the prognostic significance of CTC dynamics during the treatment of mCRC.

In the present retrospective study, we sought to investigate the correlation between CTCs and clinical outcomes (PFS, OS) for mCRC patients using an analysis of CTC trajectories up to the radiological progression of the disease.

## 2. Materials and Methods

This single-center, retrospective analysis aimed to evaluate the prognostic and predictive value of CTC trajectories for 218 patients with mCRC during first-line treatments.

Blood sampling was performed at baseline, before starting first-line treatment, and at each clinical and radiological evaluation, performed at 3-month intervals or at new symptom onset. Each patient was evaluated for the last time at radiological progression onset.

A query from our institutional medical records database was performed to identify patients affected by histologically confirmed mCRC who underwent a liquid biopsy during their first-line treatment from January 2010 to December 2013.

The inclusion criteria were: patients with measurable metastatic disease and the availability of information related to the first-line treatment performed, liquid biopsy test performed and follow-up data. Prior adjuvant treatments of any type were allowed. Patients were also stratified according to gender, primary tumor sidedness, treatment regimen and RAS mutational status (wild type (wt) vs. mutated (mut)), when available. Information on the primary tumor location was obtained from original pathology reports. Primary tumors located in the caecum, ascending colon, hepatic flexure or the proximal two-thirds transverse colon were defined as right-sided colon cancer (RCC), while tumors arising in the distal third of the transverse colon, the splenic flexure or the descending or sigmoid colon were defined as left-sided colon cancer (LCC). A rectum location was considered a standalone disease.

Follow-up was defined as the period ranging from the date of diagnosis for the first patient enrolled to the date of death for the last deceased patient. Patient data were collected using Excel 2011 (version 14.0, Microsoft Corporation, Redmont-Washington, United States of America).

The dynamic behavior of CTCs was evaluated by classifying patients according to CTC number at baseline (T0), at the first check (T1) and at the radiological progression of the disease (PD). Not detected CTCs are named CTC-ND, while a CTC number greater than 0 was named CTC+. Four groups were defined: group 1: CTC-ND (T0)/CTC-ND (T1 or PD), group 2: CTC-ND (T0)/CTC+ (T1 or PD), group 3: CTC+ (T0)/CTC-ND (T1 or PD) and group 4: CTC+ (T0)/CTC+ (T1 or PD). The corresponding trajectories for PFS and OS were evaluated and discussed.

CTC+ patients at baseline were first stratified according to the standard cut-off into 5 groups: 1, 2–3, 4–10, 11–100 and >100 CTCs/7.5 mL in order to develop a new cut-off for survival analysis. This cut-off was then used to analyze samples at T0, T1 and PD.

The prognostic value for the lower cut-off (≥1 CTCs/7.5 m) was then evaluated at baseline, T1 and PD, and a comparison between the two different cut-offs (≥1 CTCs/7.5 mL and ≥3 CTCs/7.5 mL) was performed.

### 2.1. CTC Enumeration 

From each patient, 7.5 mL of peripheral blood was collected in a CellSave tube (Menarini Silicon Biosystems) containing EDTA and a cell fixative at room temperature, and processed within 72 h. CTC enumeration was carried out with the CellSearch^®^ system using a CellSearch^®^ Epithelial Cell Kit (Menarini Silicon Biosystems), that allows the CTC enrichment through an anti-EpCAM-antibody-coated ferrofluid reagent followed by the staining for cytokeratins (CK), 4′-6-Diamidino-2-phenylindole (DAPI) and CD45. An event was considered as a CTC when having round to oval morphology, a visible nucleus, positive staining for CK and negative staining for CD45. 

### 2.2. Statistical Analysis

The clinical endpoints were PFS and OS. Categorical variables were reported as a frequency distribution, whereas continuous variables were described with the median and interquartile range (IQR) or mean value and standard deviation. Survival curves were represented with a Kaplan–Meier estimator plot and compared using a log-rank test. A *p*-value less than 0.05 was considered statistically significant. All statistical tests were 2-sided. Statistical analyses were carried out using SPSS Statistics software version 25.0 (IBM corp., Armonk, NY, USA).

### 2.3. Ethics

All patients provided written informed consent. This study was conducted according to the Good Clinical Practice guidelines and obtained approval from our institutional review board. The protocol had been previously approved by the Ethical Committee of Policlinico Umberto I (protocol n. 668/09, 9 July 2009; amended protocol 179/16, 1 March 2016).

## 3. Results

The median interval from baseline sampling to the first-time point check was 2.6 months (range: 1.5–4.2), and the median interval from baseline sampling to the radiological progression of the disease was 9 months (range: 7.7–10.3).

### 3.1. CTC Enumeration According to Patient Characteristics and Trajectories

Two hundred eighteen patients were included in this prospective study. The sample had 132 males (60%) and 86 females (40%). Half of the patients (*n* = 86, 50%) presented with LCC, while the remaining 50% was equally divided into RCC (*n* = 45, 27%) and rectal (*n* = 38, 23%) cancer. The sample was equally represented with RAS wt (*n* = 67, 50%) and RAS mut tumors (*n* = 67, 50%).

A total of 48 patients (23%) received only chemotherapy, while 145 patients (66%) and 25 patients (11%) received Bevacizumab and EGFR inhibitors (EGFRi), respectively.

Of the 218 enrolled patients, 167 (76%) had 0 CTCs at baseline (CTC-ND), while 51 (24%) were found with CTC+ at a mean of 77.68 ± 49.9 CTC number. At the time of disease progression, 83/218 (38%) patients were found with CTC+ (Table 1).

At baseline, most patients reported a CTC number < 10 (37.2% with CTCs = 1, 19.6% with CTCs = 2–3, 33%). At the time of PD, the number of patients with CTCs > 100 was not negligible (CTCs = 11–100, 15.8% and CTCs > 100, 10.5%) (Table 2).

Gender was almost equally represented in the CTC-ND and CTC+ groups.

No significant difference in CTC count was found between males and females with large standard deviations.

The highest number of CTCs was found in patients with RCC compared to LCC and rectal cancer (101.2 ± 96 vs. 4.06 ± 1.63 for LCC and 3.65 ± 1.13 for rectal cancer, *p* = 0.06).

CTC count was found to be higher in the RAS wt group compared to the RAS mut group (6.91 ± 2.19 vs. 2.4 ± 0.42, *p* = 0.034).

According to the treatments received, the highest CTC count at baseline (111.94 ± 72.37) was observed in the group of patients treated with Bevacizumab (*n* = 36, 25%), while only three patients (12%) treated with EGFRi had CTC+ at baseline (one CTC for all three patients).

In the group of patients treated with chemotherapy alone, CTCs were present at baseline in 12 patients, of whom 25% had a median number of 3.08 ± 0.92 CTCs.

In Table 1, the demographics and clinical features of the population are reported.

Group 1 (always CTC-ND) was the largest with 119 patients (55% of the total sample). Group 2 (CTC+ at PD, but CTC-ND at baseline) was composed of 48 patients (21%) that had 63 ± 47.9 CTCs. A small number of patients (16 patients, 8%) were assigned to group 3 (2.37 ± 0.65 CTCs at baseline and CTC-ND at PD). Group 4 (always CTC+) was represented by 35 patients (16%) with 86 ± 69.85 CTCs at baseline and 73 ± 38.96 CTCs at PD. Table 3 shows the CTC count for the different trajectory groups.

### 3.2. Survival Analysis According to CTC Trajectories

A summary of the analysis carried out to evaluate PFS and OS is shown in Table 4 and Table 5. The results are expressed as median and interquartile range (IQR) values in months. Median PFS and OS were 9 months (range: 7.7–10.3) and 29 months (range: 25.1–32.9) in all samples.

For CTC+ patients, PFS and OS values were evaluated by considering two different cut-offs: CTC+ (≥ 1) and CTC+ (≥ 3). The median PFS was 9 months (range 7–10.9) using ≥ 3 as cut-off, while it was 7 months (range:4.9–9.1) using ≥ 1. OS was 16 months (range 11.45–30.55) and 18 months (10.5–25.4) using ≥ 3 and ≥ 1 cut-offs, respectively. For CTC-ND patients, the median PFS and OS were 11 months (range 9.5–12.5) and 31 months (range 26.1–35.9), respectively. The comparison between CTC-ND and CTC+ (CTC = 1, CTC ≥ 1, CTC ≥ 3) groups was found to be statistically significant (*p* < 0.05).

The trajectories analysis was performed by evaluating CTC numbers at baseline sampling (T0), at T1 and at PD. In detail, group 1 (persistently CTC-ND patients) had a median PFS of 11 months at both time points (T1 and PD). Group 4 (persistently CTC+ patients) had a median PFS of 7 months (both at T1 and PD). Small differences in the values for PFS evaluated at T1 and PD were observed in the two groups (2–3), where we found a change from CTC-ND to CTC+ (group 2) or from CTC+ to TC-ND (group 3).

Group 1 showed the highest OS with median values of 36 months (range 33.7–39.3) for T1 and 34 months (range 19–38) for PD. Group 4 was characterized by lower OS with median values of 16 months (range 9.4–22.6) at T1 and 18 months (range 10.5–25) at PD.

In groups 2 and 3, the OS values were almost the same, with a small difference observed in group 2 for the values obtained at T0/T1 compared to T0/PD.

In Table 4, we report the *p*-values between the four groups. Statistical significance was found when group 1 was compared to groups 2, 3 and 4 (T0/PD analyses). Analogous results were obtained, but not reported, between group 1 and groups 2, 3 and 4 for the CTCs T0/T1 analyses. Figure 1 displays the Kaplan–Meyer survival curves (probability) for the comparison between the trajectory groups.

### 3.3. Survival Analysis According to Disease Location, Treatment Used and Mutational Status

Longer PFS and OS were found in rectal cancer, even when the differences with respect to LCC/RCC were not significant. The median PFS was 9 months (range 5–13) for RCC, 10 months (range 8.7–13.2) for LCC and 13 months (range 10.5–13.4) for rectal cancer.

OS was 26 months (range 16.8–35.6) for RCC, 31 months (range 25.6–36.4) for LCC and 40 months (range 18.5–61.5) for rectal cancer. Better outcomes were observed for CTC-ND patients, regardless of the cancer location.

Specifically, PFS was 2 months (range 6.5–17.2) for CTC-ND patients and 5 months (range 0.4–9.6) for CTC+ patients (*p* = 0.053), while OS was 30 months (range 20.7–39.7) for CTC-ND patients and 13 months (range 0.0–26.1) for CTC+ patients (*p* = 0.128). CTC-ND patients with LCC had a median PFS of 11 months (range 8.9–13.3), while CTC+ patients had a median PFS of 9 months (range 4.2–13.7). OS was 32 months (range 25.6–38.4) for CTC-ND patients and 29 months (range 19.7–38.7) for CTC+ patients. CTC-ND rectal cancer patients had a median PFS of 14 months (range 7.2–20.7), while for CTC+ patients, the median PFS was 5 months (range 0.4–9.6) (*p* = 0.010). OS was 41 months (range 20.3–61.63) for CTC-ND patients and 15 months (range 2.9–27.0) for CTC+ patients (*p* = 0.171).

The survival analysis and treatment effects on PFS and PS refer to chemotherapy alone (CHT), Bevacizumab or EGFRi in addition to CHT. The PFS values for the three categories were 7 months (range 5.1–8.8) for CHT, 11 months (range 9.4–12.5) for CHT plus Bevacizumab and 9 months (range 6.6–11.3) for CHT plus EGFRi (*p* = 0.005). The median OS was 22 months (range15.4–28.5) in the group receiving CHT alone, 31 months (range 26.1–36.2) in the group receiving CHT plus Bevacizumab and 31 months (range 18.7–43.2) in the group receiving CHT plus EGFRi (*p* = 0.003).

Patients with CTC-ND at baseline systematically yielded longer PFS and OS, and the group treated with CHT plus Bevacizumab was the only group reaching statistical significance (Table 4).

The survival analysis on RAS mutational status was not conclusive for PFS as it showed negligible differences in median values. In particular, the median PFS was 12 months (range 9.5–14.4) for RAS wt patients and 12 months (range 9.6–14.3) for RAS mut patients. On the other hand, RAS mutational status led to significant differences in OS values, showing a median OS of 36 months (range 31.1–40.8) for RAS wt compared to 26 months (range 21.4–30.5) for RAS mut (*p* = 0.014). Appendix A display the Kaplan–Meyer survival curves (probability) for different primitive cancer locations and treatments.

## 4. Discussion

The previously reported low frequency of CTCs detected with CellSearch^®^ in patients with mCRC restricts the informative potential of CTC enumeration in this disease, even in the metastatic setting, underlining the necessity for more sensitive and specific detection methods. Most studies on the prognostic role of CTC enumeration in mCRC used CTC enumeration at baseline, while—being different from other tumor types—few studies investigated the prognostic potential of longitudinal CTC quantification for mCRC to date [8,9]. In this retrospective study, we evaluated the clinical relevance of longitudinal CTC detection in 218 patients with mCRC. We confirmed a lower CTC detection rate using CellSearch^®^, compared to other solid tumors, as previously reported by different authors. Specifically, the percentage of CTCs+ patients at baseline (at least 1 CTC) was 24%, in agreement with the published literature, where the percentage of CTCs+ patients with at least one CTC was found to range between 19.5% and 35% [9] (Table 1).

At baseline, 90.2% of patients were found with fewer than 10 CTCs and 56% with fewer than 3 CTCs, although the percentage of patients with 11–100 CTCs showed an apparent increase at the disease progression, as expected.

Accordingly, Allard et al. [12] reported a mean CTC count of 4 ± 11 in mCRC, with only 17% of patients showing 5 CTCs/7.5 mL. Conversely, in metastatic breast cancer, the positivity rate at baseline has been reported to be higher, ranging from 26% to 53% according to some studies [5,11]. Similarly, in metastatic prostate cancer, the range of CTCs+ patients varies between 41% and 66%. Several possible reasons have been proposed over the years to explain the paucity of CTCs in mCRC. Among these, the hepatic filtration route, through the portal circulation, could explain the relatively low number of CTCs in this cancer compared to other carcinomas [13]. Furthermore, an additional important issue regarding the relatively low CTCs number in mCRC might be ascribed to the epithelial-to-mesenchymal transition (EMT), a process characterized by the loss of epithelial features and acquisition of mesenchymal-like markers, which is more prominent in mCRC compared to other tumor types [14]. EMT features have been associated with prognosis, where EMT-like CTCs are associated with the most aggressive tumor subtypes and the worse patient outcomes.

Since CTCs are not detectable in approximately 35% of patients, likely due to EMT, antigen-dependent CTC isolation methods have been widely questioned since they might under-represent the most aggressive and invasive CTC population [14]. In a previous study conducted in 2014 by Barbazàn et al. [15], the negative predictiveness of CTCs in mCRC was confirmed using an analysis of six EMT markers (GAPDH, VIL1, CLU, TIMP1, LOXL3 and ZEB2) in CTCs. The authors demonstrated that high expression levels of these genes on CTCs correlate with poor PFS and OS, both before and during treatment, which predicted a negative response to chemotherapy [15]. In 2015, Guinney et al. [16] classified colorectal cancer into four molecular groups, with different clinical, molecular and prognostic characteristics, namely CMS1 to CMS4. CMS1 is the most frequent type of right-sided tumor and is characterized by higher histopathological grade, high microsatellite instability, high CpG island methylator phenotype, hypermutation and BRAF mutation. Conversely, a high proportion of left-sided colon cancers are CMS4 type, showing downregulated miRNAs as miR-200 and miR-192 families, which are involved in the epithelial-to-mesenchymal transition (EMT) pathway. EMT is more frequent in left-sided cancers—as demonstrated by Nicolazzo et al. [17]—and could explain the lower CTC number found in these tumors. EMT has been widely reported to induce an underestimation of CTC number in some tumor types when using antigen-dependent assays. EMT could be one of the possible explanations for the low number of CTCs in mCRC compared to other cancer types. One way to overcome this problem could be to insert a mesenchymal marker such as vimentin in the fourth channel of CellSearch^®^ or, alternatively, to use a non-antigen-dependent method such as filtration to isolate CTCs.

A further consideration concerns the correlation between CTC count and treatment type in our patient series. In fact, despite Bevacizumab-treated patients presenting a high CTC number at baseline, most of them had no CTCs detected at the time of PD.

This last observation is consistent with that previously reported by our group in a small patient series. We hypothesized that Bevacizumab might alter CTC biological features and concluded that the lack of a predictive value for CTC counts in mCRC patients treated with Bevacizumab could represent the first example of drug-related ‘undetectable CTCs status’ using CellSearch^®^ [18].

The impact of chronic exposure to Bevacizumab on EpCAM-based detection of circulating tumor cells was further explored [19], and a decrease in the expression of EpCAM 40 kDa isoform and of cytokeratins was proved, although no evidence for EMT in treated cells was observed. The recovery rate of cells using CellSearch^®^ was gradually reduced during the course of treatment with Bevacizumab, being 84%, 70% and 40% at 1, 2 and 3 months, respectively. These results could explain the lack of CTCs at the time of disease progression in our series of Bevacizumab-treated patients.

We also investigated the correlation between CTC number and tumor location, and almost the same proportion of CTC positivity in the primary tumor situated in the rectum (21%), RCC (22%) and LCC (20%) colon were found. To the best of our knowledge, such a proportion was never investigated in the previous studies exploring the correlation of CTCs positivity with tumor location.

On the other hand, a significant difference was observed in the number of CTCs+ patients according to different anatomical locations. Patients with RCC presented a higher number of CTCs at baseline with respect to LCC/rectal cancer, confirming what was previously described in a smaller cohort [17]. The comparison between LCC and RCC cancer revealed a longer PFS and OS for LCC compared to RCC. Rectal cancer showed a further increase in PFS and OS. A direct comparison of these results with those of Cohen et al. [7] is not possible due to advances in oncological treatments since the time of the first study. Nevertheless, even if the differences are not statistically significant, the prognostic comparison between tumor side groups agrees with the recent literature [20].

Currently, it is well known that LCC and RCC are different entities of mCRC, with different origins, histology, symptoms and prognosis. In RCC and rectal cancer, we found a significant PFS difference between CTC-ND and CTC+ patients, while in LCC, the difference in PFS between CTC-ND and CTC+ patients did not reach statistical significance.

A further issue concerns the difference in CTC numbers observed between patients with RAS wt and RAS mut tumors. In particular, those with RAS mut had a lower CTC count compared to RAS wt. In our cohort, RAS mutational status did affect OS, but not PFS, independently of CTCs positivity. This result favorably compares with the current literature [20]. The PFS and OS values were not significantly different between RAS wt CTC-ND and RAS wt CTCs+ as found for RAS mut, suggesting that RAS mutational status is an independent prognostic factor. Although the prognostic role of RAS mutations is not fully elucidated to date. Unfortunately, the low number of patients who performed the RAS mutational test does not allow us to draw any significant conclusions.

We further defined four distinct groups according to CTC trajectories. More than half of the samples fell in group 1, which was characterized by patients who remained consistently CTC-ND for CTCs. The CTC count for CTC+ patients at PD was almost the same for CTC-ND/PD+ (group 2) and CTCs+/PD+ (group 4) (see Table 3). A small but interesting set is represented by patients in group 3, who had CTCs detected at baseline but not at PD. In this case, almost all patients were treated with Bevacizumab, which could be responsible for the non-detection of CTCs during the course of treatment, as previously discussed.

The proper cut-off to use for prognostic purposes in colorectal cancer patients is still a debated issue. The cut-off value for CTC enumeration using CellSearch^®^ has been validated by the FDA. In the specific case of metastatic colorectal cancer, the established cut-off was three CTCs/7.5 mL of blood. Consequently, the cut-off of three was not arbitrarily set. Concerning the cut-off of 1 CTC, we had previously evaluated the prognostic and predictive significance of CTC count at baseline and under treatment in 119 mCRC subjects and compared the standard cutoff (≥3 CTCs/7.5 mL to ≥1 CTCs/7.5 mL) [10]. The presence of at least one CTC at baseline count was found predictive for poor prognosis, suggesting that patients with even one CTC should be switched from the favorable prognostic group—conventionally defined by the presence of fewer than three CTCs—to the unfavorable, deserving more careful monitoring.

Using a cut-off of ≥ one CTCs/7.5 mL, four prognostic trajectories were investigated. The best outcome was reported in patients of group 0 (CTC-ND at T0, T1 and PD). In this group, the median PFS and OS were significantly longer compared to all other groups. This result agrees with Cohen et al. [7], who found that patients starting with favorable conditions (CTC-ND) and always maintaining this status, had the best outcomes in terms of PFS and OS. On the other hand, the comparison between groups 2, 3 and 4 showed no remarkable or statistically significant differences in PFS and OS. Moreover, in two previous studies, Tol et al. [21] and Sastre et al. [22] reported that the CTC baseline count defines three different prognostic trends. Specifically, patients who are always CTC-ND have the longest PFS and OS; patients with a high CTC count at one time point have the shortest survival outcomes; and those with a CTC count that converted from CTC+ to CTC-ND have intermediate survival outcomes. In a meta-analysis completed in 2014, Huang et al. [9] reported that CTCs detected during treatment were significantly associated with the response rate and disease control rate (*p* < 0.05). In the author’s opinion, the results confirm the prognostic independent role of CTC count but also underline possible technical issues in CTCs enumeration using CellSearch^®^ for mCRC patients during the course of treatment.

## 5. Conclusions

The present results demonstrate that for mCRC patients, the value of baseline CTC enumeration has a limited impact on PFS and OS, in agreement with a similar study published by Tol et al. [21], while CTC trajectories evaluated at PD represent better prognostic indicators. Defining CTC trajectories using a serial CTC assessment may help to improve risk stratification, providing potential biomarkers to monitor treatment response in mCRC. Moreover, the sample homogeneity in the first line of treatment as well as the sampling at disease progression represent innovative characteristics of the present study compared to previous studies reported in the literature

## Figures and Tables

**Figure 1 cells-12-01172-f001:**
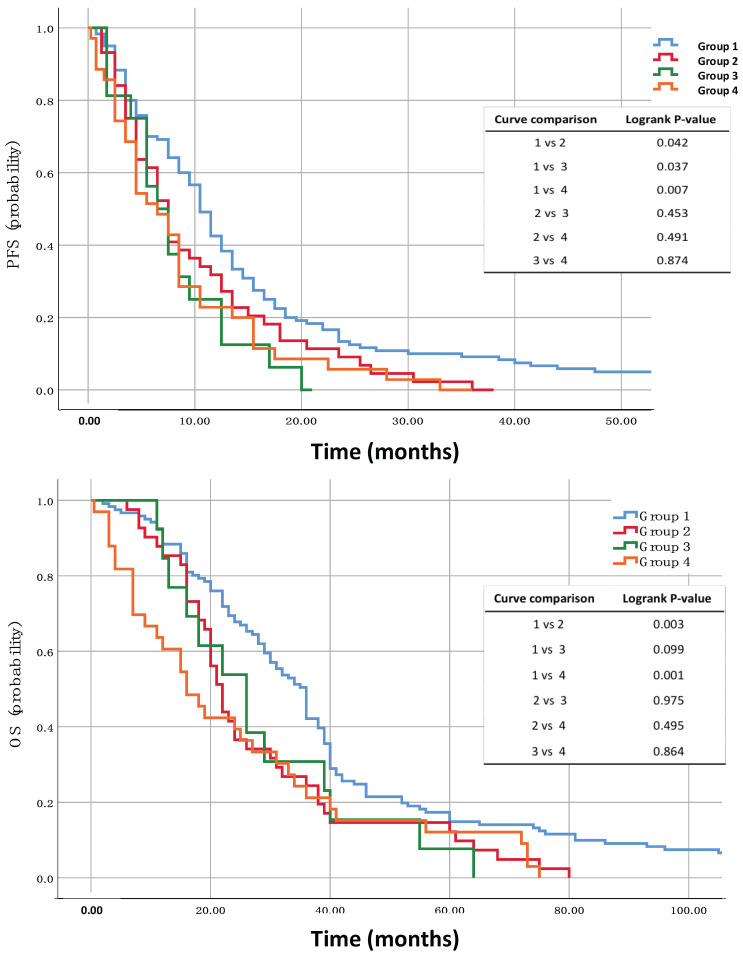
Kaplan–Maier survival curves (probability) for PFS (top) and OS (bottom). Comparison between the 4 trajectory groups. Group 1 (119 patients) CTC-ND (baseline)/CTC-ND (progression disease). Group 2 (48 patients): CTC-ND (baseline)/CTC+ (progression disease). Group 3 (16 patients): CTC+ (baseline)/CTC-ND (progression disease). Group 4 (35 patients): CTC+ (baseline)/CTC+ (progression disease).

**Table 1 cells-12-01172-t001:** Patient demographics and clinical characteristics. CTC count was evaluated at baseline as CTCs > 0 and reported as mean values ± standard deviation. Percentage values for CTC+ and CTC-ND were evaluated with respect to the total number in the subgroup.

	Patients *N* (%)	CTC-ND *N* (%)	CTC+ *N* (%)	CTCs Count (Baseline)	
All Patient	218 (100)	167 (76)	51 (24)	77.68 ± 49.9	
Gender					
Female	86 (40)	68 (79)	18 (21)	136.6 ± 130.7	*p* = 0.38
Male	132 (60)	99 (75)	33 (25)	45.5 ± 31.3	
Type of cancer					
RCC	45 (27)	35 (78)	10 (22)	101.2 ± 96	*p* = 0.06
LCC	86 (50)	69 (80)	17 (20)	4.06 ± 1.63	
Rectal	38 (23)	30 (79)	8 (21)	3.65 ± 1.13	
Not available	49				
Mutational status					
RAS mut	67 (50)	52 (78)	15 (22)	2.4 ± 0.42	*p* = 0.034
RAS wt	67 (50)	55 (82)	12 (18)	6.91 ± 2.19	
Ras not available	84				
First-line treatment					
CHT + Bevacizumab	145 (66)	109 (75)	36 (25)	111.94 ± 72.37	*p* = 0.634
CHT + EGFRi	25 (11)	22 (88)	3 (12)	1 ± 0	
CHT only	48 (23)	36 (75)	12 (25)	3.08 ± 0.92	

CTCs: circulating tumor cells; ND: not detected; LCC: left-sided colon cancer; RCC: right colon cancer; mut: mutated; wt: wild type; CHT: chemotherapy; EGFRi: EGFR inhibitors.

**Table 2 cells-12-01172-t002:** Patients with CTC+ at baseline and PD. Distribution versus different CTC count groups.

	*N* (%) Baseline	*N* (%) PD
CTCs = 1	19 (37.2)	28 (33.3)
CTCs = 2–3	10 (19.6)	19 (22.8)
CTCs = 4–10	17 (33.4)	14 (17.5)
CTCs = 11–100	3 (5.8)	13 (15.8)
CTCs > 100	2 (3.9)	9 (10.5)

CTCs: circulating tumor cells; PD: progression disease.

**Table 3 cells-12-01172-t003:** CTC trajectories. Patient number and mean CTC count (mean ± standard deviation) for different conditions at baseline and at the progression of the disease.

CTCs Trajectories	*N* (%)	CTC (Baseline)	CTC (PD)
CTC-ND (Baseline)/CTC-ND (PD)	119 (55)	0	0
CTC-ND (Baseline)/CTC+ (PD)	48 (21)	0	63 ± 47.9
CTC+(Baseline)/CTC-ND (PD)	16 (8)	2.37 ± 0.65	0
CTC+ (Baseline)/CTC+ (PD)	35 (16)	86 ± 69.85	73 ± 38.96

CTC: circulating tumor cell; ND: not detected; PD: progression disease.

**Table 4 cells-12-01172-t004:** PFS and OS data. CTC trajectories evaluated at baseline (T0) and at the first check (T1) or at PD.

	PFS (Months)	*p*-Value	OS (Months)	*p*-Value
All Patients	9 (7.7–10.3)		29 (25.1–32.9)	
CTC-ND	11 (9.5–12.5)	<0.05 ^Note 1^	31 (26.1–35.9)	<0.05 ^Note 1^
CTC+ (=1) (T0)	8 (6.59–6.41)		25 (14.24–36.77)	
CTC+ (≥3) (T0)	9 (7–10.9)		16 (11.45–30.55)	
CTC+ (≥1) (T0)	7 (4.9–9.1)		18 (10.5–25.4)	
CTCS Trajectories				
Group 1: CTC-ND (T0)/CTC-ND (PD) CTC-ND (T0)/CTC-ND (T1)	11 (9.6–12.4) 11 (9.5–17.5)	^Note 2^	36 (33.7–39.3) 34 (19–38)	^Note 3^
Group 2: CTC-ND (T0)/CTC+ (≥1) (PD) CTC-ND (T0)/CTC+ (≥1) (T1)	8 (6.6–9.4) 9 (2–15)		22 (19.5–24.5) 31 (26–36)	
Group 3: CTC+ (≥1) (T0)/CTC-ND (PD) CTC+ (≥1) (T0)/CTC-ND (T1)	7 (4.7–9.2) 8 (5.8–10.1)		26 (16.8–35.2) 24 (16.7–31.2)	
Group 4: CTC+ (≥1) (T0)/CTC+ (≥1) (PD) CTC+ (≥1) (T0)/CTC+ (≥1) (T1)	7 (4.6–9.0) 7 (4.9–9.0)		16 (9.4–22.6) 18 (10.5- 25)	

CTCs: circulating tumor cells; ND: not detectable; PD: progression disease, PFS: progression-free survival; OS: overall survival. Note 1. *p*-value < 0.05 refers to the comparison between CTC-ND and CTC+ groups (CTCs = 1, CTCs ≥ 1, CTCs ≥ 3). Note 2. *p*-value for groups 1 vs. 2: *p* = 0.042; 1 vs. 3: *p* = 0.037; 1 vs. 4: *p* = 0.007; 2 vs. 3: *p* = 0.453; 2 vs. 4: *p* = 0.491; 3 vs. 4: *p* = 0.874. Note 3. *p*-value for groups. 1 vs. 2: *p* = 0.003; 1 vs. 3: *p* = 0.099; 1 vs. 4: *p* = 0.001; 2 vs. 3: *p* = 0.975; 2 vs. 4: *p* = 0.495; 3 vs. 4: *p* = 0.864.

**Table 5 cells-12-01172-t005:** PFS and OS data. Analysis concerning disease location, the treatment used and mutational status. CTCs were evaluated at baseline (T0).

	PFS (Months)	*p*-Value	OS (Months)	*p*-Value
Disease Location				
LCC	10 (8.7–13.2)	^Note 1^	31 (25.6–36.4)	^Note 2^
RCC	9 (5–13)		26 (16.8–35.6)	
Rectal cancer	13 (10.5–13.4)		40 (18.5–61.5)	
LCC CTC-ND	11 (8.9–13.3)	0.762	32 (25.6–38.4)	0.520
LCC CTC+ (≥1)	9 (4.2–13.7)		29 (19.7–38.7)	
RCC CTC-ND	12 (6.5–17.2)	0.053	30 (20.7–39.7)	0.128
RCC CTC + (≥1)	5 (0.4–9.6)		13 (0.0–26.1)	
Rectal cancer CTC-ND	14 (7.2- 20.7)	0.010	41 (20.3–61.63)	0.171
Rectal cancer CTC+ ( ≥ 1)	5 (0.4–9.6)		15 (2.9–27.0)	
Treatment				
CHT	7 (5.1–8.8)	0.005	22 (15.4–28.5)	0.003
CHT+BEV	11 (9.4–12.5)		31 (26.1–36.2)	
CHT+EGFRi	9 (66–11.3)		31 (18.7–43.2)	
CHT/CTC-ND	7 (4.1–9.2)	0.873	22 (16.3–27.7)	0.875
CHT/CTC+ (≥1)	6 (2.6–9.4)		18 (3.9–32.1)	
CHT+BEV/CTC-ND	12 (10.1–13.4)	0.003	35 (30.1–39.6)	0.002
CHT+BEV/CTC+ (≥1)	8 (6.1–9.9)		19 (10.9–27.1)	
CHT+EGFRi/CTC-ND	9 (7.1–10.9)	0.029	31 (19.1–42.5)	0.645
CHT+EGFRi/CTC+ (≥1)	6 (0.5–14.0)		3 (0–4)	
RAS mutation				
RAS wt	12 (9.5–14.4)	0.310	36 (31.1–40.8)	0.014
RAS mut	12 (9.6–14.3)		26 (21.4–30.5)	
RAS wt/CTC-ND	13 (10.4–15.5)	0.389	36 (30.9–41.1)	0.047
RAS mut/CTC-ND	12 (10.8–13.2)		27 (22.1–31.9)	
RAS wt/CTC+ (≥1)	9 (7.8–10.1)	0.557	36 (23.8–48.1)	0.008
RAS mut/CTC+ (≥1)	9 (2.7–15.2)		26 (13.6–38.6)	

CTC: circulating tumor cell; ND: not detected; PD: progression disease, PFS: progression-free survival; OS: overall survival, LCC: left-sided colon cancer; RCC: right colon cancer; ChT: chemotherapy; Bev: Bevacizumab; wt: wild type; mut: mutated. Note 1. *p*-value for groups. LCC vs. RCC: *p* = 0.39; LCC vs. Rectal: *p* = 0.22; RCC vs. Rectal: *p* = 0.102 Note 2. *p*-value for groups. LCC vs. RCC: *p* = 0.814; LCC vs. Rectal: *p* = 0.271; RCC vs. Rectal: *p* = 0.233.

## Data Availability

Data will be shared by the corresponding author upon reasonable request.

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
