# Peer review of "Prognostic Role of Circulating Tumor Cell Trajectories in Metastatic Colorectal Cancer"

_cells, 2023, doi:10.3390/cells12081172_

Round 1
Reviewer 1 Report
Manuscript ID cells-2288931
Title: Prognostic Role of Circulating Tumor Cells Trajectories in metastatic colorectal cancer.
Authors: Valentina Magri * , Luca Marino , Chiara Nicolazzo , Angela Gradilone , Gianluigi De Renzi , Michela De Meo , Orietta Gandini , Arianna Sabatini , Daniele Santini , Enrico Cortesi , Paola Gazzaniga
Manuscript describes clinical study evaluating circulating tumor cells (CTCs) applicability as a prognostic liquid biopsy marker detected in patients diagnosed with metastatic colorectal cancer (mCRC). The concept is not a novel one, as CellSearch method was reported in a vast number of studies, including mCRC. This work can be of interest to some readers as liquid biopsies and biomarkers isolated are an important concept in realizing precision medicine.
Manuscript can be published after the following comments and questions are addressed:
1. The main critique relates to lack of data for the control experiments with healthy donors. Ultimately, the cutoff for the CTC should be determined in sync with the data collected for healthy donors (i.e., how many epithelial cells detected in blood of HD resembled the phenotype of the CTC?). The specificity of the assay, sensitivities for different cohorts, ROC are not reported.
2. The cutoff for the CTC in mCRC maybe debated in literature, but ultimately there should be bioanalytical figures of merits presented that discuss rate of false positives, false negatives, and clearly justify use of 1 or 3 CTC as a cutoff.
3. Why arbitrarily the cutoff was set to 1 or 3 CTC? Why not higher?
4. How clinically useful really is isolation of CTC based on EpCAM antigens exclusively, if the authors note shortcomings due to EMT and other factors causing EpCAM to significantly decrease? What other marker could be useful?
5. In light of authors discussion on EpCAM downregulation on CTC surface, either due to treatment or EMT, I suggest change the wording from CTC negative to CTC “not detected.”
6. Was viability of CTCs assessed in patients undergoing different treatments?
7. There is a real abundance of KM curves in the manuscript and same data in tables. I propose being more selective in presenting these in the main manuscript.
8. Figure captions for KM: please provide sample size.
9. In the discussion, the authors use LCC and RCC but acronym is not explained. Reviewer assumes these refer to the left colon cancer and right colon cancer, but it is not consistent with the text in Table (i.e., left side cancer, right side cancer). Please clarify and edit.
Page3, lines 103 - 106 contain wording that appears verbatim in line 123-127. Remove one.
Reviewer 2 Report
This manuscript is about a study of prognosis of metastatic colorectal cancer through enumeration of circulating tumor cells (CTCs) from the patients’ peripheral blood samples. This is a very important research topic and very useful for guiding a clinical practice by the cancer liquid biopsy. Although there are many reports about analysis of CTCs for monitoring the progression of the tumor and evaluation of the clinical treatment, more clinical research data about the correlation of the CTCs numbers in the blood sample and Progression-free-survival (PFS) and overall survival are still necessary for knowing the prognostic role of CTCs, especially using FDA cleared Cell Search system. Despite colorful techniques for capturing and identifying CTCs from clinical blood samples, a reliable relation between the CTCs number and carcinoma status is hard to be judged. So, this study was well-designed and manuscript has been well-organized. In this retrospective study, the dynamic change of the number of CTCs before and after treatment as baseline and ongoing status of more than 200 mCRC patients may provide more reasonable predictive value for investigating the patients’ clinical staging and survival. The experiment results upon a cohort of patients exhibited an obviously informative potential of enumeration of single CTCs as a biomarker to predict the PFS and OS of patients. However, some concerns haven’t been reached due to the restriction of the EpCAM mediated cancer cells targeting strategy, and lack of adequate gene sequencing information of the RAS mutation, which can be further studied in future.
There are some typos or format errors in this manuscript, the authors need to modify them carefully. Such as: “be line with” should be “be in line with”; the first character of the word “Bevacizumab” should be consistent in the format of upper or lower letter in the context.
Round 2
Reviewer 1 Report
Accept for publication with one small edit.
Please include the answer to my first question (related to controls, i.e., CTCs detected in healthy donors) into the manuscript. I do not believe I found that in the edited manuscript.
The performance of the system is extensively described in 2004 by Allard et al. (2004)[12]. Prevalence of CTC was determined in blood from healthy donors. The control population contained practically no CTC (average=XXX, range =XXX, n=XXX) in 7.5 mL.
Author Response
Thank you for your further suggestion. We added in the manuscript the answer to your first question (written in red), with the numerical details, as requested.
We hope that the revised version of the manuscript satisfies the requirements for pubblication.
best regards
Valentina Magri